# ToTRL: Unlock LLM Tree-of-Thoughts Reasoning Potential through Puzzles Solving

## Abstract

Large language models (LLMs) demonstrate significant reasoning capabilities, particularly through long chain-of-thought (CoT) processes, which can be elicited by reinforcement learning (RL). However, prolonged CoT reasoning presents limitations, primarily verbose outputs due to excessive introspection. The reasoning process in these LLMs often appears to follow a trial-and-error methodology rather than a systematic, logical deduction. In contrast, tree-of-thoughts (ToT) offers a conceptually more advanced approach by modeling reasoning as an exploration within a tree structure. This reasoning structure facilitates the parallel generation and evaluation of multiple reasoning branches, allowing for the active identification, assessment, and pruning of unproductive paths. This process can potentially lead to improved performance and reduced token costs. Building upon the long CoT capability of LLMs, we introduce tree-of-thoughts RL (ToTRL), a novel on-policy RL framework with a rule-based reward. ToTRL is designed to guide LLMs in developing the parallel ToT strategy based on the sequential CoT strategy. Furthermore, we employ LLMs as players in a puzzle game during the ToTRL training process. Solving puzzle games inherently necessitates exploring interdependent choices and managing multiple constraints, which requires the construction and exploration of a thought tree, providing challenging tasks for cultivating the ToT reasoning capability. Our empirical evaluations demonstrate that our ToTQwen3-8B model, trained with our ToTRL, achieves significant improvement in performance and reasoning efficiency on complex reasoning tasks.

## 1 Introduction

Large language models (LLMs) have recently demonstrated remarkable capabilities in tackling complex reasoning tasks, with many advanced LLMs (Anthropic, 2025; Guo et al., 2025; Du et al., 2025; Google DeepMind, 2025) commonly employing the chain-of-thought (CoT) (Wei et al., 2022) technique to generate explicit, step-by-step intermediate reasoning. This sequential reasoning process has enabled powerful inference in structured domains like mathematics and programming. For example, the GPT-o1 series (OpenAI, 2024) achieves improved inference performance with longer reasoning chains. Similarly, DeepSeek-R1 (Guo et al., 2025) attains notable results in complex reasoning via the emergent long CoT reasoning process, which is activated using reinforcement learning (RL) with a rule-based reward signal.

However, the fundamental nature of CoT is a linear and single-path reasoning process. Although effective for certain problems, this sequential structure inherently limits its efficiency when addressing tasks that necessitate exploring and evaluating multiple potential solution trajectories. A critical issue arises when using RL with rule-based rewards to induce longer reasoning, the resulting outputs can become verbose and redundant (OpenAI, 2024; Guo et al., 2025; Du et al., 2025). This is because the underlying reasoning remains local, moving from one step to the next without a global perspective or an effective mechanism to evaluate the overall promise of a path or prune unpromising lines of thought. This contrasts with the efficient human cognitive strategy, which involves considering alternatives and focusing resources globally.

In contrast, the tree-of-thoughts (ToT) method (Yao et al., 2023) offers a conceptually superior approach by explicitly modeling the reasoning process as an exploration across a tree structure of potential thoughts or states. The tree-based reasoning structure facilitates the parallel generation and

evaluation of diverse reasoning branches, enabling the LLM to actively identify, evaluate, and prune unproductive thought paths. By maintaining a global view of the search space, ToT holds the potential to achieve higher performance and significantly reduce redundant exploration and associated token costs compared to the linear CoT reasoning process.

Building upon the capabilities of long CoT (Qwen Team, 2025), we introduce tree-of-thoughts RL (ToTRL), a novel on-policy RL framework using the rule-based reward. ToTRL is designed to guide LLMs in developing the parallel ToT strategy (Yao et al., 2023) based on the sequential CoT strategy (Yeo et al., 2025). Directly building ToT reasoning based on CoT within the reasoning mode is challenging due to the established habituation to the sequential CoT style. To address this, ToTRL employs a two-stage training strategy. Initially, the LLM is trained to perform ToT reasoning in a non-thinking mode, leveraging more moldable thinking patterns to activate ToT reasoning. Once the LLM has developed a degree of ToT reasoning ability in the non-reasoning mode, it undergoes further training in the reasoning mode. This second stage aims to enable the LLM to effectively utilize its newly acquired ToT capabilities during inference on complex tasks, building upon its existing CoT reasoning strengths.

Moreover, we employ LLMs as players in puzzle games that require tree-based reasoning, providing challenging tasks for cultivating the ToT reasoning capability. These puzzle games are deliberately chosen for their intrinsic requirement of exploring interdependent choices and managing multiple constraints simultaneously, requiring a thought tree construction and exploration involving multiple concurrent hypotheses and future states.

Our contributions are summarized as follows:

- Introduce ToTRL, a novel on-policy RL framework using the rule-based reward for developing the LLMs' ToT reasoning strategy based on the long CoT capability.
- Employ LLMs as players in puzzle games that require tree-based reasoning, providing challenging tasks for cultivating the ToT reasoning capability.
- Provide empirical evaluations showing that our ToTQwen3-8B, trained with ToTRL, achieves significant performance and reasoning efficiency on complex tasks.

## 2 TREE-OF-THOUGHTS RL

Building upon the capabilities of long CoT (Qwen Team, 2025), we introduce tree-of-thoughts RL using the on-policy RL strategy (Section 2.1) with the rule-based reward (Section 2.2). Specifically, ToTRL employs a two-stage training strategy (Section 2.3) and serves LLMs as the puzzle game player for training (Section 2.4).

### 2.1 ON-POLICY RL ALGORITHM

Test-time scaling introduces a significant paradigm shift for LLMs (OpenAI, 2024; Guo et al., 2025). This approach enables long CoT reasoning and fosters sophisticated reasoning behaviors, leading to superior performance on complex reasoning tasks. A key technique facilitating these advancements is rule-based RL, which elicits behaviors such as self-verification and iterative refinement.

In this paper, we employ the on-policy RL algorithm (Schulman et al., 2017; Zhang et al., 2021; Hu, 2025; Ahmadian et al., 2024) for our proposed ToTRL method. Specifically, for each prompt $q$, we sample $n$ responses $\{o_1, o_2, \ldots, o_n\}$ from the old policy $\pi_{\theta_{\text{old}}}$, where $n$ is the number of sampled trajectories (i.e., the rollout size per prompt). The policy model $\pi_\theta$ is then optimized by maximizing the following surrogate objective:

$$\mathcal{J}(\theta) = \mathbb{E}_{q \sim \mathcal{D}, \{o_i\}_{i=1}^n \sim \pi_{\theta_{\text{old}}}(o|q)}$$

$$\frac{1}{n} \sum_{i=1}^n \left( \min\left( \frac{\pi_\theta(o_i|q)}{\pi_{\theta_{\text{old}}}(o_i|q)} A_i, \text{clip}\left( \frac{\pi_\theta(o_i|q)}{\pi_{\theta_{\text{old}}}(o_i|q)}, 1 - \epsilon, 1 + \epsilon \right) A_i \right) - \beta \cdot \mathbb{D}_{\text{KL}}(\pi_\theta || \pi_{\text{ref}}) \right), \quad (1)$$

where $\epsilon$ and $\beta$ are hyperparameters. The advantage estimate $A_i$ is computed based on the rule-based reward function, using the sampled rewards $\{r_1, r_2, \ldots, r_n\}$, and is calculated as:

$$A_i = r_i - \frac{1}{n-1} \sum_{j \neq i} r_j. \quad (2)$$

**ToT Guidance**

**System:** You are a helpful assistant. The user asks questions and the assistant answers them. The assistant first thinks with **tree-of-thoughts reasoning** and then provides the answer to the user. The tree-of-thoughts reasoning and answer are contained in the **<tot> </tot>** and **<answer> </answer>** tags, respectively, that is, **<tot> The tree-of-thoughts reasoning goes here </tot> <answer> The answer goes here </answer>**.

**User Input**

**User:** Solve the following Alphametic puzzle: **EEM+EGYM=EYME**. In this puzzle, each letter represents a unique digit (0-9). The same letter always represents the same digit, and different letters represent different digits. Leading digits cannot be zero. Find ALL solutions that make the equation true. You are required to solve the problem using the tree-of-thoughts reasoning. **Please perform tree-of-thoughts reasoning step by step and don't skip any intermediate steps.** Show your tree-of-thoughts process within **<tot> </tot>** tags. List ALL solutions using json format within **<answer> </answer>** tags with no duplicates.

**Stage1: No-thinking Mode**

**Assistant:**
**<think>**
**</think>**

**<tot>**
## ToT Reasoning Process Here
**</tot>**
**<Answer>**
{"E": 6, "M": 3, "G": 0, "Y": 7}, {"E": 4, "M": 2, "G": 3, "Y": 8}, {"E": 2, "M": 1, "G": 6, "Y": 9}, {"E": 2, "M": 6, "G": 1, "Y": 3}
**</Answer>**

**Stage2: Thinking Mode**

**Assistant:**
**<think>**
## ToT Reasoning Process Here
**</think>**
**<tot>**
## Summarization of Reasoning Process
**</tot>**
**<Answer>**
{"E": 6, "M": 3, "G": 0, "Y": 7}, {"E": 4, "M": 2, "G": 3, "Y": 8}, {"E": 2, "M": 1, "G": 6, "Y": 9}, {"E": 2, "M": 6, "G": 1, "Y": 3}
**</Answer>**

Figure 1: Overview of multi-stage ToT guidance with solving an Alphametic puzzle as an example.

Traditionally, RL algorithms incorporate a KL divergence penalty to regulate the divergence between the online policy model and the frozen reference model (Yu et al., 2025a). However, during training with ToTRL, the model distribution can diverge significantly from the initial model. The KL penalty term will restrict the exploration of model outputs. Consequently, we exclude the KL term by setting $\beta = 0$.

## 2.2 REWARD MODELING

To effectively shape the LLM's learning trajectory via reinforcement learning, we designed a rule-based reward function $R(o|q; r_{\text{s}}, r_{\text{p}})$. This function employs a strict hierarchical evaluation protocol, prioritizing format validity before evaluating correctness. The reward mechanism is parameterized by two key constants, including the reward magnitude $r_{\text{s}}$ for a correct output and the penalty magnitude $r_{\text{p}}$ assigned for any detected errors.

**Format Validity.** The initial validation phase rigorously examines whether the generated output $o$ conforms to all $K$ predefined structural and syntactic constraints $C = \{c_1, c_2, \ldots, c_K\}$. A binary indicator function $v_k(o)$ for the outcome of each individual check $c_k$ can be defined as:

$$v_k(o) = \begin{cases} 1, & \text{if check } c_k \text{ passes for output } o, \\ 0, & \text{otherwise.} \end{cases} \tag{3}$$

An output $o$ is deemed format-valid if and only if it satisfies all $K$ checks. Consequently, the format validity indicator, $\mathcal{F}(o)$ can be formally defined as:

$$\mathcal{F}(o) = \mathbb{I}(\sum_{k=1}^{K} v_k(o) = K). \tag{4}$$

If $\mathcal{F}(o) = 0$, the evaluation terminates immediately, assigning the penalty reward $r_{\text{p}}$.

**Accuracy Evaluation.** Subsequent evaluation of correctness occurs strictly conditional upon successful format validation. This stage evaluates whether the structurally valid output $o$, which is expected to contain a sequence representing potentially multiple solutions, accurately represents the complete set of ground truth solutions $Y(x)$. Let $S(o)$ denote the set of solutions extracted from the model's output sequence $o$ The accuracy indicator $\mathcal{A}(o|q)$, based on the equality between the extracted set of solutions and the ground truth set, can be formulated as:

$$\mathcal{A}(o|x) = \mathbb{I}(S(o) = Y(x)), \tag{5}$$

where $S(o) \equiv Y(x)$ holds if and only if $S(o)$ contains all solutions in $Y(x)$ and no others.

**Rule-based Reward.** The reward function $R(o|q; r_{\text{s}}, r_{\text{p}})$ integrates the outcomes of the format validity and accuracy evaluation, parameterized by $r_{\text{s}}$ and $r_{\text{p}}$. The reward is calculated based on the

format validity status $\mathcal{F}(o)$ and the accuracy status $\mathcal{A}(o|x)$, which can be calculated:

$$R(o|q; r_{\mathrm{s}}, r_{\mathrm{p}}) = \begin{cases} r_{\mathrm{p}}, & \text{if } \mathcal{F}(o) = 0, \\ \mathcal{A}(o|x) \cdot r_{\mathrm{s}} + (1 - \mathcal{A}(o|x)) \cdot r_{\mathrm{p}}, & \text{if } \mathcal{F}(o) = 1. \end{cases} \tag{6}$$

Alternatively, the total reward can be expressed as:

$$R(o|x; r_{\mathrm{s}}, r_{\mathrm{p}}) = \mathcal{F}(o)\mathcal{A}(o|x)(r_{\mathrm{s}} - r_{\mathrm{p}}) + r_{\mathrm{p}}. \tag{7}$$

This compact form highlights that the reward is fundamentally the base penalty $r_{\mathrm{p}}$, potentially incremented by the difference $(r_{\mathrm{s}} - r_{\mathrm{p}})$ only when both format and accuracy indicators ($\mathcal{F}(o)$ and $\mathcal{A}(o|x)$) are simultaneously active. In practice, we provide a clear binary success signal for ToTRL, and the reward parameters are instantiated with $r_{\mathrm{s}} = 1$ and $r_{\mathrm{p}} = -1$.

## 2.3 MULTI-STAGE TOTRL

Conventional CoT reasoning enforces a strictly linear, step-by-step reasoning process (Wei et al., 2022; Yeo et al., 2025). This linearity can be restrictive when exploring multiple potential solution paths. Inspired by ToT (Yao et al., 2023), which structures problem-solving as a deliberate exploration of a thought tree, ToTRL adapts this concept to enhance LLM reasoning within an RL setting. The original ToT framework represents the reasoning process as a rooted tree, where each node corresponds to an intermediate thought $s$. From a state $s_{d-1}$ at depth $d-1$, a generator $G(\cdot)$ proposes potential next thoughts, forming a set of candidate states at depth $d$:

$$T_d \supseteq \{G(s) \mid s \in T_{d-1}\}, \tag{8}$$

where $T_0 = \{S_0\}$ represents the initial problem state. Conceptually, ToT involves evaluating these thoughts using a value function $V(s)$ and pruning less promising branches to manage the search space effectively.

In our ToTRL approach, instead of implementing an explicit external search algorithm, we develop a ToT guidance prompt to facilitate the parallel generation of ToT reasoning processes within each RL rollout. As depicted in Figure 1, LLMs are required to solve problems by employing ToT reasoning, ensuring each step is executed sequentially without omitting any intermediate stages. This prompt-driven process encourages the LLM to explore ToT reasoning trajectories autonomously within its own generation process. The output generated after ToT reasoning is subsequently evaluated using Equation (6), and the entire process is optimized via Equation (1).

However, existing reasoning LLMs are primarily accustomed to the sequential CoT reasoning style (Qwen Team, 2025). Consequently, integrating ToT reasoning directly into current CoT-based reasoning LLMs presents a significant challenge. To address this challenge, ToTRL employs a multi-stage ToT guidance strategy. Initially, as illustrated in Figure 1, the LLM undergoes training to perform ToT reasoning in a non-thinking mode. The non-reasoning mode is achieved by introducing blanks between reasoning tags, which compels the model to suspend its conventional reasoning processes. Once the LLM demonstrates an initial proficiency in ToT reasoning within the non-reasoning mode, it proceeds to further training in the reasoning mode. This subsequent stage aims to enable the LLM to develop its newly acquired ToT capabilities based on the established CoT reasoning strengths.

## 2.4 LLM AS PUZZLE GAME PLAYER

During the training process with our ToTRL, LLMs are employed as players in puzzle games. These games necessitate tree-based reasoning, thereby providing challenging tasks for cultivating the LLMs' ToT reasoning capabilities. They are specifically selected due to their inherent demand for exploring interdependent choices and managing multiple concurrent constraints, which inherently requires the construction and exploration of a sophisticated thought tree involving numerous concurrent hypotheses and future states. Specifically, as shown in Figure 2, we leverage Sudoku and Alphametic puzzles during the training process with ToTRL for this purpose.

**Sudoku Puzzle.** Sudoku puzzle is an ideal game for cultivating ToT reasoning. The high interdependency of choices in Sudoku puzzles means that placing a single digit significantly impacts other cells, necessitating a forward-looking evaluation of cascading implications. Critically, the need for

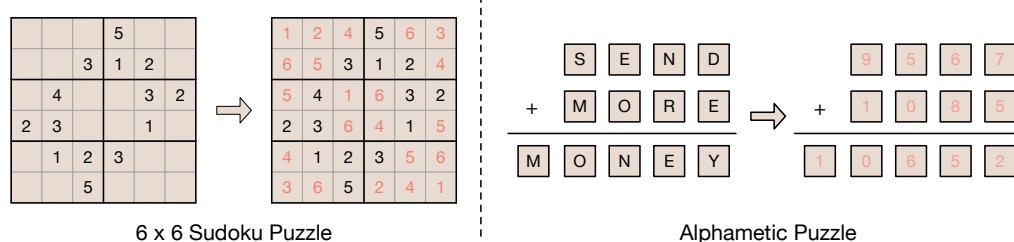

Figure 2: During the training process with ToTRL, LLMs are employed as players in puzzle games, including Sudoku and Alphametic puzzles.

hypothetical reasoning and backtracking directly corresponds to the construction and exploration of a thought tree, where decision nodes represent potential branches that explore their consequences.

**Alphametic Puzzle.** Alphametic puzzles also offer an ideal environment for cultivating ToT reasoning. They exhibit strong interdependency due to the critical role of carry-overs. The primary challenge lies in simultaneously satisfying both the mathematical correctness of the equation and the combinatorial constraint of assigning a unique digit to each letter, presenting a complex, dual-constraint environment. The iterative process of hypothetical assignments, consequence propagation, and backtracking upon contradiction inherently forms a tree-based search.

Employing LLMs as players in puzzle games, we provide challenging environments designed to foster their tree-based reasoning capabilities within complex constraint satisfaction problems.

## 3 EXPERIMENTS

### 3.1 EXPERIMENT SETTINGS

**Training Data.** The ToTQwen3-8B model is trained using 1440 puzzle games, specifically designed to cultivate its ToT reasoning capabilities. We provide more details in Appendix B.1.

**Implementation Details.** We train the ToTQwen3-8B model using our ToTRL framework, initializing it from the Qwen3-8B model (Qwen Team, 2025). The training process utilized the AdamW optimizer (Loshchilov & Hutter, 2017) with a constant learning rate of $1 \times 10^{-6}$, incorporating a warm-up ratio of 0.01. We employed a batch size of 9, a rollout size of 16, and a maximum sequence length of 16384 tokens, with no weight decay applied. The model undergoes full-parameter fine-tuning for one epoch using DeepSpeed-Zero stage 2 with CPU offload (Rajbhandari et al., 2020), distributed across four 80 GB A100 GPUs.

**Baselines.** We select the existing SOTA reasoning LLMs with similar parameters as baselines of our ToTQwen3-8B model, which include DeepSeek-R1-Distill-Qwen-7B (Guo et al., 2025), Llama-3.1-Nemotron-Nano-8B (Bercovich et al., 2025), GLM-4-Z1-9B-0414 (ZhiPuAI, 2025), Phi-4 Reasoning (Abdin et al., 2025), and Qwen3-8B (Qwen Team, 2025). Notably, Qwen3-8B and ToTQwen3-8B use the thinking mode for evaluation.

**Evaluation Benchmarks.** To comprehensively evaluate the proficiency of our ToTQwen3-8B model, we utilize a set of logic reasoning tasks categorized as either in-distribution or out-of-distribution (OOD). In-distribution tasks, included in the training data, comprise 6×6 Sudoku and Alphametic puzzles. OOD tasks, which are not part of the training data, include 5×5 Cross-word (Yao et al., 2023), 9×9 Sudoku (bryanpark, 2017), K&K puzzles (Xie et al., 2025), Poker 24 Game, and Make 24 puzzles. Additionally, we also leverage the widely adopted AIME 2024–2025 (American Invitational Mathematics Examination) and AMC 2023 (American Mathematics Competitions) benchmarks, both known for their rigorous and diverse mathematical problems. Specifically, for the evaluation of our ToTQwen3-8B, we employ the ToT prompt (Figure 1) for logic reasoning tasks while leveraging the CoT prompt for mathematical problems. For all evaluation tasks, we use accuracy as the performance metric. Notably, the accuracy is calculated on the number of correct answers for puzzles with multiple solutions.

Table 1: Performance of ToTQwen3-8B on in-distribution puzzle solving tasks.

| | 6×6 Sudoku | Alphametic Puzzle | | | | |
|---|---|---|---|---|---|---|
| | | 1 sol | 2 sol | 3 sol | 4 sol | Avg. |
| DeepSeek-R1-Distill-Qwen-7B | 0.000 | 0.020 | 0.030 | 0.020 | 0.040 | 0.028 |
| Llama-3.1-Nemotron-Nano-8B | 0.000 | 0.100 | 0.100 | 0.127 | 0.160 | 0.122 |
| GLM-4-Z1-9B-0414 | 0.200 | 0.080 | 0.140 | 0.500 | 0.290 | 0.253 |
| Phi-4 Reasoning | 0.120 | 0.420 | 0.680 | 0.640 | 0.645 | 0.596 |
| Qwen3-8B (Thinking) | 0.660 | 0.820 | 0.980 | 0.960 | **0.960** | 0.930 |
| **ToTQwen3-8B (Ours)** | **0.800** | **0.960** | **1.000** | **0.973** | **0.960** | **0.973** |

Table 2: Performance of ToTQwen3-8B on OOD logic reasoning tasks with a unique solution.

| | 5×5 Crossword | 9×9 Sudoku | K&K Puzzle |
|---|---|---|---|
| DeepSeek-R1-Distill-Qwen-7B | 0.000 | 0.000 | 0.007 |
| Llama-3.1-Nemotron-Nano-8B | 0.002 | 0.000 | 0.043 |
| GLM-4-Z1-9B-0414 | 0.062 | 0.000 | 0.893 |
| Phi-4 Reasoning | 0.000 | 0.080 | 0.957 |
| Qwen3-8B (Thinking) | 0.378 | 0.180 | 0.950 |
| **ToTQwen3-8B (Ours)** | **0.508** | **0.260** | **0.986** |

**Thinking Budget.** Thinking budget facilitates control over the thinking process through manual interruption. Specifically, when the LLM's thinking duration reaches a predefined threshold, the process is halted by inserting a stop instruction. Subsequently, the LLM generates a final response based on the partial reasoning accumulated. Notably, this budget was adjusted based on task complexity. A smaller budget of 8K is allocated for the simple K&K Puzzle, while a larger budget of 32K is used for the more complex 9×9 Sudoku. We provide more details about thinking budget in Appendix B.3.

### 3.2 In-Distribution Logic Reasoning

As shown in Table 1, performance on these logic tasks varies across the evaluated models. Notably, the ToTQwen3-8B model demonstrates significantly superior performance in in-distribution tasks compared to the other evaluated models on both tasks. Specifically, ToTQwen3-8B achieves the highest success rate on the 6×6 Sudoku task. On the Alphametic Puzzles, it consistently performs at a high level across puzzles with varying numbers of solutions.

ToTQwen3-8B is built upon the Qwen3-8B model. A comparison of their performance reveals that ToTQwen3-8B shows a substantial improvement over Qwen3-8B, which scored 0.660 on Sudoku and averaged 0.930 on Alphametic puzzles. This significant performance gain is attributed to specialized training on these in-distribution tasks and the ToT reasoning strategy integration within our ToTQwen3-8B. The ToT reasoning strategy enables the LLM to engage in more global reasoning by exploring multiple paths, which is particularly effective for finding single or multiple solutions in complex constraint satisfaction problems.

### 3.3 Out-of-Distribution Logic Reasoning

Besides in-distribution tasks, we also evaluate our ToTQwen3-8B model on OOD logic reasoning tasks. As shown in Table 2 and Table 3, many models exhibit very low scores (close to 0), indicating a significant struggle with these logic reasoning tasks.

**Logic Reasoning with Unique Solution.** Table 2 presents performance on OOD logic reasoning tasks, all of which have a single correct solution. It is important to note that for tasks with unique solutions, some models might occasionally arrive at the correct answer through guessing rather than a complete logical derivation process. Our ToTQwen3-8B consistently achieves the highest scores across all three unique solution tasks evaluated, 0.508 on 5×5 Crossword, 0.260 on 9×9 Sudoku, and 0.986 on K&K Puzzle. This consistent, high performance on diverse OOD tasks provides strong

Table 3: Performance of ToTQwen3-8B on OOD logic reasoning tasks with multiple solutions.

| | Poker 24 Game | | | | | Make 24 Puzzle | | | | |
|---|---|---|---|---|---|---|---|---|---|---|
| | 1 sol | 2 sol | 3 sol | 4 sol | Avg. | 1 sol | 2 sol | 3 sol | 4 sol | Avg. |
| DeepSeek-R1-Distill-Qwen-7B | 0.025 | 0.025 | 0.017 | 0.019 | 0.021 | 0.050 | 0.025 | 0.000 | 0.000 | 0.019 |
| Llama-3.1-Nemotron-Nano-8B | 0.100 | 0.075 | 0.042 | 0.050 | 0.067 | 0.050 | 0.088 | 0.058 | 0.031 | 0.057 |
| GLM-4-Z1-9B-0414 | 0.625 | 0.500 | 0.208 | 0.306 | 0.410 | 0.650 | 0.625 | 0.575 | 0.460 | 0.578 |
| Phi-4 Reasoning | 0.175 | 0.113 | 0.092 | 0.038 | 0.104 | 0.250 | 0.275 | 0.400 | **0.488** | 0.353 |
| Qwen3-8B (Thinking) | 0.700 | 0.463 | 0.342 | 0.369 | 0.468 | 0.750 | 0.600 | 0.625 | 0.481 | 0.614 |
| **ToTQwen3-8B (Ours)** | **0.900** | **0.713** | **0.392** | **0.406** | **0.603** | **0.850** | **0.638** | **0.675** | 0.481 | **0.661** |

Table 4: Performance of ToTQwen3-8B on OOD mathematical tasks.

| | AIME 2024 | AIME 2025 | AMC 2023 | Avg. |
|---|---|---|---|---|
| DeepSeek-R1-Distill-Qwen-7B | 0.533 | 0.367 | 0.925 | 0.608 |
| Llama-3.1-Nemotron-Nano-8B | 0.600 | 0.367 | 0.900 | 0.623 |
| GLM-4-Z1-9B-0414 | **0.667** | 0.600 | **0.950** | 0.739 |
| Phi-4 Reasoning | **0.667** | 0.467 | 0.925 | 0.686 |
| Qwen3-8B (Thinking) | 0.633 | 0.533 | 0.900 | 0.689 |
| **ToTQwen3-8B (Ours)** | **0.667** | **0.633** | **0.950** | **0.750** |

evidence that our ToTQwen3-8B model effectively leverages the underlying logical constraints and employs a well-developed derivation process, rather than relying on chance.

**Logic Reasoning with Multiple Solutions.** Table 3 presents performance on more OOD logic reasoning tasks, including Poker 24 Game and Make 24 Puzzle, which admit multiple valid solutions. The primary challenge lies not only in identifying a valid solution but also potentially in finding multiple distinct solutions or performing robustly across puzzles with varying numbers of possible solutions. This necessitates a reasoning process capable of exploring diverse successful paths rather than merely converging on a single outcome. As shown in Table 3, ToTQwen3-8B again outperforms all other models on both the Poker 24 Game and Make 24 Puzzle, demonstrating superior performance across puzzles with varying numbers of solutions. This advantage is most pronounced on puzzles with many solutions, where the capacity to explore multiple valid reasoning paths is essential. The ToT strategy inherently explores a tree of possibilities, branching at critical decision points. This systematic exploration of multiple potential continuations provides a global perspective, making it inherently well-suited for tasks admitting diverse valid solutions. Unlike long CoT reasoning strategy that commits to a single path, ToT can explore parallel branches, significantly increasing the probability of discovering multiple distinct solutions when they exist.

In summary, from an OOD perspective, the results presented in Table 2 and Table 3 strongly indicate that ToTQwen3-8B possesses superior generalization capabilities on these logic reasoning tasks. Its robust performance on both unique and multiple-solution puzzles suggests that the ToT reasoning structure confers a significant advantage in addressing unfamiliar logic challenges by facilitating a more comprehensive exploration of reasoning possibilities, a capability that is particularly valuable for tasks requiring the discovery of multiple valid solutions.

### 3.4 OUT-OF-DISTRIBUTION MATHEMATICAL TASKS

We further investigate whether our ToTRL can enhance complex reasoning abilities can transfer to a highly challenging mathematical reasoning scenario. According to Table 4, ToTQwen3-8B demonstrates strong performance on these OOD mathematical reasoning tasks. It achieves the highest average score of 0.750 across the three benchmarks. Specifically, ToTQwen3-8B performs comparably well on AIME 2024 with a score of 0.667, matching the performance of GLM-4-Z1-9B-0414 (ZhiPuAI, 2025) and Phi-4 Reasoning (Abdin et al., 2025) with more model parameters. On AIME 2025, ToTQwen3-8B leads with a score of 0.633, outperforming all other listed models. For the AMC 2023 benchmark, ToTQwen3-8B achieves a score of 0.950, which is tied with GLM-4-Z1-9B-0414 (Abdin et al., 2025) for the highest performance.

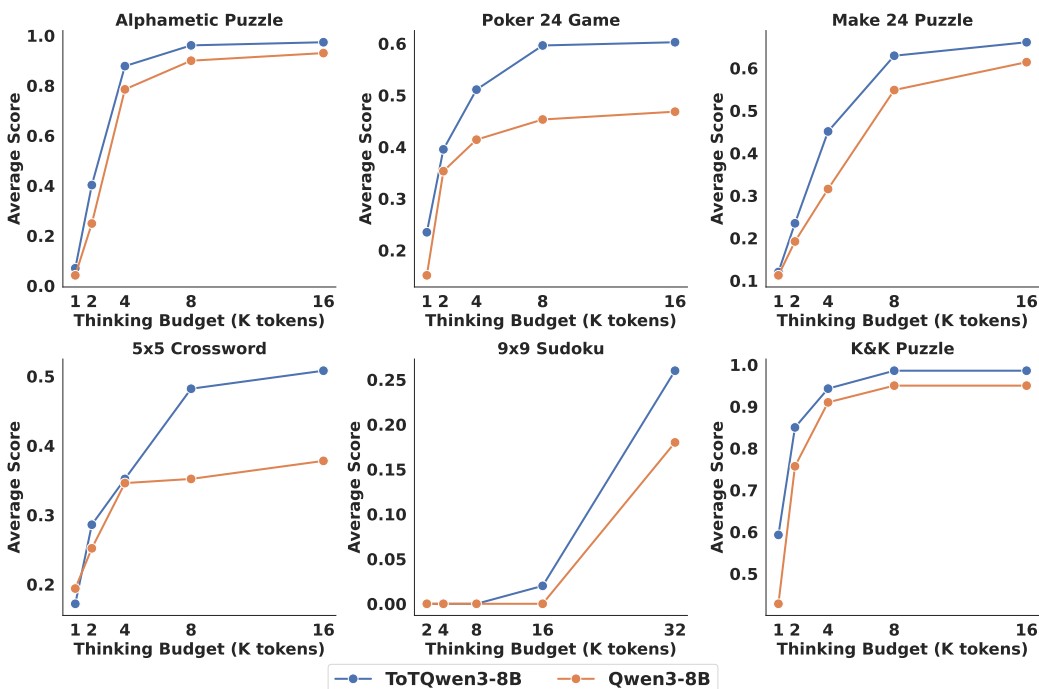

Figure 3: Illustration of test time scaling on logic reasoning tasks.

The ToT training process encourages the LLM to generate more diverse, parallel, and structured intermediate reasoning steps, with each parallel branch employing the CoT format. This approach ultimately improves the quality of the sequential reasoning steps generated within the CoT prompt, which is crucial for tackling complex OOD problems. The strong performance observed on OOD tasks indicates that our ToTRL framework effectively enhances the LLM's reasoning capabilities. Importantly, these improved abilities generalize well to novel and challenging mathematical problems beyond the training distribution, suggesting that ToTRL successfully cultivates a more robust and transferable form of reasoning.

## 3.5 TEST TIME SCALING

Test-time scaling (Yu et al., 2025b) is an innovative approach in language modeling that leverages additional computational resources during the testing phase to enhance performance. This method has shown significant promise in various domains, including language modeling and code generation (OpenAI, 2024; Guo et al., 2025; Du et al., 2025; Google DeepMind, 2025). We investigate the test time scaling of our proposed ToTQwen3-8B model against a baseline Qwen3-8B (Qwen Team, 2025) model using different thinking budgets across six logic reasoning tasks.

As illustrated in Figure 3, the performance of both ToTQwen3-8B and Qwen3-8B generally improves with an increased thinking budget across all tasks. This indicates that allowing more computational resources for intermediate thinking steps leads to better reasoning outcomes. Importantly, Figure 3 demonstrates the efficiency of the ToTQwen3-8B approach, which leverages the ToT reasoning strategy. ToTQwen3-8B consistently outperforms Qwen3-8B across various thinking budgets. Notably, ToTQwen3-8B is often able to reach a higher average score with a smaller thinking budget compared to Qwen3-8B. This suggests that the structured tree-of-thoughts reasoning employed by ToTQwen3-8B allows it to explore the solution space more effectively and efficiently, requiring fewer tokens (and thus less computational cost and time) to achieve superior or comparable performance. This efficiency is a key advantage, making ToTQwen3-8B a more practical solution for logic reasoning tasks under computational constraints.

Table 5: Ablation study on ToT guidance.

| | CoT | ToT | Crossword 5×5 | Sudoku 9×9 | K&K Puzzle | Poker 24 Game | Make 24 Puzzle |
|---|---|---|---|---|---|---|---|
| Qwen3-8B (Enable-Thinking) | ✔ | | 0.378 | 0.180 | 0.950 | 0.468 | 0.614 |
| | | ✔ | 0.376 | 0.080 | 0.700 | 0.485 | 0.600 |
| ToTQwen3-8B | ✔ | | 0.350 | 0.200 | 0.971 | 0.519 | 0.648 |
| | | ✔ | **0.508** | **0.260** | **0.986** | **0.603** | **0.661** |

Table 6: Ablation study on multi-stage ToTRL.

| | Stage 1 | Stage 2 | Crossword 5×5 | Sudoku 9×9 | K&K Puzzle | Poker 24 Game | Make 24 Puzzle |
|---|---|---|---|---|---|---|---|
| Qwen3-8B | | ✔ | 0.470 | 0.160 | 0.957 | 0.485 | 0.532 |
| | ✔ | ✔ | **0.508** | **0.260** | **0.986** | **0.603** | **0.661** |

## 3.6 ABLATION STUDIES

To further investigate the contribution of ToT guidance prompts and the multi-stage training process within ToTRL, we perform a series of ablation studies. Notably, additional ablation studies are provide in Appendix C.

**ToT Guidance.** Table 5 presents an ablation study on the effectiveness of ToT guidance compared to CoT guidance. For the Qwen3-8B model, employing ToT guidance yields mixed results. The performance improves on some tasks but decreases significantly on others. This suggests that although ToT guidance holds potential, the Qwen3-8B model is not adept at utilizing the ToT reasoning strategy. In contrast, when applying ToT guidance to the ToTQwen3-8B model, we observe consistent and significant improvement across all reported tasks for both guidance types. This clearly demonstrates that ToT guidance is substantially more effective than CoT, particularly when paired with a model specifically trained to leverage its tree-like exploration capabilities. Notably, ToTQwen3-8B also demonstrates improved performance with CoT guidance, indicating that ToTRL can also facilitate CoT reasoning capabilities.

**Multi-Stage ToTRL.** Table 6 presents an investigation into the contribution of the multi-stage training process within ToTRL, comparing models trained with different stages. The results demonstrate that the inclusion of Stage 1 training significantly improves performance across all evaluated tasks. Specifically, Stage 1 training focuses on enabling the LLM to perform the fundamental steps of ToT reasoning using no-thinking mode, thereby facilitating the integration of this capability into models primarily accustomed to sequential CoT reasoning. This indicates that the initial exploration and tree-building capabilities acquired during Stage 1 are crucial for effective ToT guidance and the superior performance on diverse reasoning tasks.

## 4 CONCLUSION

In this work, we introduced ToTRL to guide LLMs from sequential CoT reasoning to a more efficient ToT reasoning strategy. By employing a two-stage training process and utilizing puzzle games that necessitate tree-based exploration, we successfully cultivate ToT capabilities in LLMs. Our ToTQwen3-8B, trained with ToTRL, demonstrates substantially superior performance on in-domain logic puzzles and generalization to OOD tasks. Furthermore, these enhanced reasoning abilities transfer effectively to challenging mathematical benchmarks. Crucially, our model also exhibits greater efficiency, achieving higher scores with smaller thinking budgets during test-time scaling compared to its CoT-based counterpart. These results collectively overcome the verbosity and local perspective limitations of long CoT, showcasing the robust potential and practical advantages of explicit tree-based reasoning for advanced AI on diverse complex tasks.

## DECLARATION OF LLM USAGE

The usage of LLMs is strictly limited to aid and polish the paper writing.

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

# A RELATED WORKS

## A.1 TREE-OF-THOUGHTS

Early applications of ToT (Yao et al., 2023; Long, 2023) approaches relied on external search algorithms (e.g., BFS, DFS) or auxiliary modules (e.g., prompter agents, checkers, memory modules, and ToT controllers) to manage planning, decision-making, and backtracking. Inspired by AlphaZero, TS-LLM (Feng et al., 2023) proposed learning a dedicated value function to guide the tree search and iteratively improving the LLM itself, aiming to handle deeper and more complex search trees. Furthermore, ATS (Zhang et al., 2023) allows LLMs to perform complex tree-search reasoning by generating the entire search trajectory in a single response, which can be activated by prompting LLMs. Collectively, these efforts demonstrate the significant potential of internalizing ToT capabilities within the LLM itself, moving towards more autonomous reasoning. However, significant challenges remain in achieving truly autonomous planning and decision-making. Existing methods frequently depend on external search control and prompting rather than developing the LLM's inherent capacity for tree-based reasoning. Consequently, building upon the long-CoT reasoning capability, we develop ToTRL to train LLMs to perform ToT reasoning.

## A.2 LONG-COT ACTIVATED THROUGH RL

RL (Schulman et al., 2017; Shao et al., 2024; Ahmadian et al., 2024) has proven an effective approach for eliciting longer CoT, thereby enhancing LLMs' reasoning capabilities. Recent studies (Guo et al., 2025; Du et al., 2025) have demonstrated that LLMs can acquire and extend their reasoning paths, including reflection and verification, using RL with simple rule-based rewards. Data-driven RL research has further broadened the application of this approach. For instance, Logic-RL (Xie et al., 2025) utilized logic puzzles for training, demonstrating generalization to mathematics, while SWE-RL (Wei et al., 2025) leveraged software evolution data to improve model performance on software engineering tasks and OOD reasoning. However, despite somewhat enhancing the depth of thought and problem-solving capabilities, current RL-elicited long CoT (Wei et al., 2022; Long, 2023) reasoning paradigms are inherently linear, following a single exploration path. This sequential structure is inefficient for tackling complex problems that necessitate extensive exploration and evaluation of multiple potential solutions. Lacking a global perspective and effective mechanisms for path evaluation and pruning, this approach often generates redundant outputs and incurs unnecessary computational overhead. In this paper, we develop ToTRL to guide LLM reasoning from a global perspective, which aims to improve performance while reducing token costs.

# B EXPERIMENT DETAILS

Table 7: Training Dataset Size Breakdown by Stage and Task

| Task | Stage 1 Samples | Stage 2 Samples | Total Samples |
|---|---|---|---|
| 6×6 Sudoku | 540 | 180 | 720 |
| Alphametic Puzzle | 540 | 180 | 720 |
| **Total per stage** | 1080 | 360 | 1440 |

## B.1 TRAINING DATA

As detailed in Table 7, the training data is distributed between two types of self-generated puzzles, including 720 6×6 Sudoku puzzles and 720 Alphametic puzzles. The training regimen is divided into two stages, with stage 1 comprising 1080 puzzles and stage 2 comprising 360 puzzles. Notably, the Alphametic puzzles in the training set are specifically designed to feature between 1 and 4 unique solutions, thereby ensuring the model's exposure to a diverse range of problem complexities.

## B.2 EVALUATION BENCHMARKS

The performance of the ToTQwen3-8B is rigorously evaluated across various tasks, encompassing both in-distribution puzzles and OOD challenges that the model has not encountered previously. A

Table 8: Overview of Evaluation Benchmarks

| | Dataset Source | Test Data Size | Samples/Sol. |
|---|---|---|---|
| **In-Distribution** | | | |
| 6×6 Sudoku | Self-generated Dataset | 50 | - |
| Alphametic Puzzle | Self-generated Dataset | 200 | 50 |
| **Out-of-Distribution** | | | |
| 5×5 Crossword | ToT Dataset (Yao et al., 2023) | 50 | - |
| 9×9 Sudoku | Kaggle Sudoku Dataset (bryanpark, 2017) | 50 | - |
| K&K Puzzle | K&K Puzzles Dataset (Xie et al., 2024) | 140 | 20 |
| Poker 24 Puzzle | Self-generated Dataset | 160 | 40 |
| Make 24 Puzzle | Self-generated Dataset | 160 | 40 |

Table 9: Ablation study on ToT thinking across various LLMs.

| | Crossword 5×5 | Sudoku 9×9 | K&K Puzzle | Poker 24 Game | Make 24 Puzzle |
|---|---|---|---|---|---|
| DeepSeek-R1-Distill-Qwen-7B | 0.000 | 0.000 | 0.012 | 0.010 | 0.000 |
| Llama-3.1-Nemotron-Nano-8B | 0.000 | 0.000 | 0.032 | 0.047 | 0.033 |
| GLM-4-Z1-9B-0414 | 0.060 | 0.000 | 0.710 | 0.356 | 0.417 |
| Phi-4 Reasoning | 0.000 | 0.040 | 0.722 | 0.054 | 0.232 |
| Qwen3-8B (Thinking) | 0.376 | 0.080 | 0.700 | 0.485 | 0.600 |
| **ToTQwen3-8B (Ours)** | **0.508** | **0.260** | **0.986** | **0.603** | **0.661** |

comprehensive overview of these evaluation benchmarks, including their categories, sources, test set sizes, and specific sample distributions, is presented in Table 8.

For in-distribution evaluation, the evaluation dataset for 50 6×6 Sudoku puzzles and 200 Alphametic puzzles is self-generated. The Alphametic Puzzle test set maintains a distribution of 50 puzzles per solution count category, including 1 to 4 solutions, to robustly assess performance.

The OOD evaluation dataset is derived from established datasets and self-generated puzzles. Standardized puzzles include 50 5×5 Crossword puzzles from the ToT dataset (Yao et al., 2023) and 50 9×9 Sudoku puzzles randomly selected from a Kaggle Dataset (bryanpark, 2017) to test generalization to more complex Sudoku formats. The K&K Puzzle benchmark, sourced from the original K&K Puzzles dataset (Xie et al., 2024), comprises 140 test puzzles, specifically chosen to provide 20 puzzles for each of its 7 distinct solution categories. To further probe multi-solution reasoning in novel contexts, we self-generated 160 puzzles each for the Poker 24 Puzzle and Make 24 Puzzle. These datasets are carefully structured to include 40 samples for problems featuring 1-4 distinct solutions, allowing for a nuanced evaluation of the model's ability to identify multiple valid outcomes.

### B.3 THINKING BUDGET

The thinking budget for the model's intermediate thinking output is predefined (Qwen Team, 2025). Should the thinking output reach the thinking budget, the thinking process is terminated, and a standardized instruction is immediately introduced: "Considering the limited time by the user, I have to give the solution based on the thinking directly now." The model then generates its final response based on the reasoning accumulated up to the termination point. This procedure guarantees that all methods compared operate under an equivalent and constrained computational budget.

## C SUPPLEMENTARY ABLATION STUDIES

**Thinking with ToT.** As indicated in Table 9, applying the ToT reasoning strategy via prompting to the untrained baseline models resulted in performance that is generally inferior or highly inconsistent compared to their native CoT reasoning. This finding strongly validates the core motivation of this work: untrained LLMs cannot effectively utilize the ToT strategy through simple prompting alone. Specialized training with ToTRL is essential to unlock and maximize this potential.

Table 10: Ablation study on reward modeling.

| | Reward Modeling | Crossword 5×5 | Sudoku 9×9 | K&K Puzzle | Poker 24 Game | Make 24 Puzzle |
|---|---|---|---|---|---|---|
| Qwen3-8B | Partial-Credit | 0.200 | 0.040 | 0.836 | 0.238 | 0.315 |
| | Full-Credit | **0.508** | **0.260** | **0.986** | **0.603** | **0.661** |

Table 11: Ablation study on different training paradigms.

| | Training Paradigm | Crossword 5×5 | Sudoku 9×9 | K&K Puzzle | Poker 24 Game | Make 24 Puzzle |
|---|---|---|---|---|---|---|
| Qwen3-8B | CoT | 0.404 | 0.180 | 0.957 | 0.526 | 0.628 |
| | ToT | **0.508** | **0.260** | **0.986** | **0.603** | **0.661** |

**Reward Modeling.** Table 10 presents an exploration of partial-credit rewards within the ToTRL framework. As shown, employing partial rewards caused the model to prematurely converge on sub-optimal strategies that only output a subset of the correct answers. This ultimately collapsed the intended ToT thinking process and yielded lower overall scores compared to the full-credit approach.

**Training Paradigms.** Table 11 presents an ablation study comparing the effectiveness of ToT guidance against CoT guidance under an identical total number of training steps. As demonstrated, the ToT thinking paradigm provides a clear performance improvement over the CoT paradigm. Consequently, the observed gain cannot be attributed merely to extended training.

## D  CASE STUDIES

We analyze the following alphametic puzzle: $QQB + QVQ = VBG$, where each letter represents a distinct digit and leading zeros are disallowed. The objective is to enumerate all possible assignments that satisfy this equation. Figure 4 illustrates the reasoning trajectory and final solutions produced by the Qwen3-8B using a standard CoT prompt, while Figure 5 displays the tree-structured reasoning process of ToTQwen3-8B under a ToT prompt. We utilize this specific example to compare the behavior of these two distinct reasoning approaches on the same task.

The two approaches exhibit notable differences in their reasoning structure. As shown in Figure 4, the CoT model develops its reasoning along a single, linear path. It documents local column-wise addition and carry constraints while repeatedly alternating between different hypotheses for the values of Q and V within that same path. Consequently, the distinct search branches become entangled, lacking clear case partitioning. This structural deficiency makes checks across branches prone to redundancy and omissions. In contrast, ToTQwen3-8B in Figure 5 initially partitions the possible values of Q into several cases based on the constraint in the hundreds column, and then further divides these into subcases according to the value of V. Within each resulting branch, the model sequentially applies the constraints from the units, tens, and hundreds columns. As soon as a contradiction emerges in any column, that branch is immediately pruned and is not expanded further, yielding a structurally clear search tree that contains substantially less redundant exploration.

The disparity in reasoning structure is further evident in the quality of the outputs. In Figure 4, the CoT model's "<answer>" block provides three assignments. Although $(Q, V, B, G) = (1, 2, 3, 4)$ and $(2, 4, 6, 8)$ satisfy all column-wise addition and digit-uniqueness constraints, the assignment $(2, 4, 7, 9)$ is inconsistent with the global constraints yet is incorrectly labeled as another correct solution. Furthermore, the valid solution $(4, 9, 3, 7)$ is entirely missed. This demonstrates that, in the linear CoT mode, the model's global verification is unreliable, especially in multi-solution settings. Conversely, in Figure 5, the ToT model explicitly differentiates valid solutions from no-solution branches throughout the tree and ultimately aggregates the three correct assignments: $(1, 2, 3, 4)$, $(2, 4, 6, 8)$, and $(4, 9, 3, 7)$. These assignments precisely constitute all true solutions to the puzzle, with no missing, incorrect, or duplicate entries.

Table 12: Comparison to explicitly guided search algorithms on poker 24 game.

| Game | CoT | ToT | TS-LLM | ToTRL |
|------|-----|-----|--------|-------|
| Poker 24 Game | 0.468 | 0.631 | 0.582 | 0.603 |

Table 13: Performance of ToTQwen3-8B on OOD real-world tasks.

| Game | BFCL v3 (Live) | LiveCodeBench | Arena Hard | Creative Writing v3 |
|------|----------------|---------------|------------|---------------------|
| Qwen3-8B (Thinking) | 0.767 | **0.556** | 0.832 | 0.727 |
| ToTQwen3-8B (Ours) | **0.783** | 0.554 | **0.858** | **0.756** |

## E    DISCUSSION

### E.1    ToT ACTIVATION

Similar to how long CoT reasoning process is often activated and generalized via RL training on mathematical tasks (Guo et al., 2025), our contribution demonstrates that an appropriate task set (specifically, puzzles) can effectively activate the ToT thinking mode. As indicated in Table 13, we observe modest performance improvements on agentic tool use (BFCL v3 (Yan et al., 2024)) and writing tasks (Arena Hard (Li et al., 2024), Creative Writing (Samuel J. Paech, 2024)), while no statistically significant difference has been observed so far on coding benchmarks (LiveCodeBench v5 (Jain et al., 2024)). These findings suggest that our approach holds preliminary yet promising potential for real-world applications. Future work can focus squarely on generalizing this learned exploration policy using RL to real-world scenarios.

### E.2    EXPLICITLY-GUIDED SEARCH STRATEGIES

Explicitly guided search algorithms typically necessitate customized frameworks for different tasks. In contrast, our method avoids explicitly guiding the model to use a specific search algorithm (e.g., Breadth-First Search (BFS) or Monte Carlo Tree Search (MCTS)); instead, it trains a policy that learns to explore. To demonstrate the performance disparity between these approaches, we compare ToTRL with ToT (Yao et al., 2023) and TS-LLM (Feng et al., 2023) on the Poker 24 Game, utilizing the Qwen3-8B-think model for both ToT and TS-LLM. As shown in Table 12, our ToTRL achieves performance similar to these explicitly guided tree-search methods, but offers greater practicality. This improved practicality stems from requiring only a single invocation and eliminating the need for a task-specific framework. This result effectively demonstrates the strong utility of internalizing the ToT thinking mode into LLMs.

## LIMITATIONS

Despite the promising results of ToTQwen3-8B, this work has several limitations. The transferability of ToTQwen3-8B to different reasoning tasks requires a thorough investigation. Furthermore, our approach implicitly induces ToT reasoning through guided prompting and a two-stage reinforcement learning strategy, leveraging training data based on long CoT outputs. This method aims to encourage a more global problem perspective and potentially reduce computational overhead compared to generating exhaustive CoT traces for every thought path. However, a limitation is that the model, trained on these CoT foundations, may still produce partially redundant outputs. This redundancy can stem from the underlying long CoT structure, even with the ToT mechanism attempting to focus exploration. This implicit induction, although effective in partially mitigating CoT habituation and demonstrating the benefits mentioned above, may exhibit different exploratory dynamics and optimality compared to explicit tree search algorithms. Explicit algorithms could potentially offer stricter pruning or different trade-offs in managing redundancy.

**User:** Solve the following alphametic puzzle: QQB+QVQ=VBG. In this puzzle, each letter represents a unique digit (0-9). The same letter always represents the same digit, and different letters represent different digits. Leading digits cannot be zero. Find ALL solutions that make the equation true.\nList ALL solutions using json format within <answer> </answer> tags with no duplicates.

**Assistant:**
```
<think>
Q=1. Then V can be 2 or 3.
Q=1, V=2 ... -> Q=1, V=2, B=3, G=4 -> 113 + 121 = 234 -> **valid solution.**
Q=1, V=3 ... -> contradiction -> **no solution here.**
Wait, are there any other possibilities for Q=1, V=3, etc. Let's double-check Q=1 thoroughly.
...
Q=2. Then V can be 4 or 5.
...
From the equations we get that **B can be 6 or 7**.
First, B=6:
   ... -> 226 + 242 = 468.
  VBG is 4 (V), 6 (B), 8 (G) -> 468. **Valid solution.**
Next, B=7:
   ... -> 227 + 242 = 469.
  VBG is 4 (V), 7 (B), 9 (G) -> 469. **Correct. Another solution.**
...
But wait! double check Q=2.
...
Q=3 ... -> contradictions -> **no solution here.**
Wait, did we miss any for Q=3? Let's double-check.
...
**So total solutions so far are:**
1. Q=1, V=2, B=3, G=4 -> 113 + 121 = 234
2. Q=2, V=4, B=6, G=8 -> 226 + 242 = 468
3. Q=2, V=4, B=7, G=9 -> 227 + 242 = 469
**These are all unique. I don't see any duplicates. So the final answer should include these three solutions.**
</think>

<Answer>
{"Q": 1, "V": 2, "B": 3, "G": 4}, {"Q": 2, "V": 4, "B": 6, "G": 8}, {"Q": 2, "V": 4, "B": 7, "G": 9}
</Answer>
```

Figure 4: Illustration of CoT reasoning pattern with solving an Alphametic puzzle as an example.

```
User: Solve the following Alphametic puzzle: QQB+QVQ=VBG.\nIn this puzzle, each letter represents a unique
digit(0-9). The same letter always represents the same digit, and different letters represent different digits.
Leading digits cannot be zero. Find ALL solutions that make the equation true. You are required to solve the
problem using the tree-of-thoughts reasoning. Please perform tree-of-thoughts reasoning step by step and don't
skip any intermediate steps. Show your tree-of-thoughts process within <tot> </tot> tags. List ALL solutions
using json format within <answer> </answer> tags with no duplicates.
```

```
Assistant:
<think>
Case 1: Q = 1.
  V can be 2 or 3.
    Subcase 1a: Q=1, V=2….
    Subcase 1b: Q=1, V=3 -> ... -> **no solution.**
**So for Q=1, only Subcase 1a gives a valid solution: Q=1, V=2, B=3, G=4.**
Case 2: Q = 2.
  V = 4 or 5.
    Subcase 2a: Q=2, V=4.
      ... tens place -> B can be 6 or 7.
      - If B=6 -> Q=2, V=4, B=6, G=8 -> 226 + 242 = 468 -> valid solution.
      - If B=7 -> ... units place gives G = -1 -> invalid.
    Subcase 2b: Q=2, V=5 -> ... -> **no solution here.**
**So for Q=2, only Subcase 2a gives a solution, and only B=6 is valid: Q=2, V=4, B=6, G=8.**
...
Case 4: Q = 4.
  V = 8 or 9.
    Subcase 4a: Q=4, V=8 -> ... -> **no solution here.**
    Subcase 4b: Q=4, V=9.
      ... tens place -> B can be 3 or 4.
      - If B=3 -> Q=4, V=9, B=3, G=7 -> 443 + 494 = 937 -> valid solution.
      - If B=4 -> ... -> invalid.
        **Thus, only B=3 is valid here.**
**So Subcase 4b gives one solution: Q=4, V=9, B=3, G=7.**
**So, the solutions found are:**
1. Q=1, V=2, B=3, G=4
2. Q=2, V=4, B=6, G=8
3. Q=4, V=9, B=3, G=7
...
**Yes, no duplicates in any of them. Therefore, these are the three solutions.**
</think>
<tot>
To solve the alphametic puzzle QQB + QVQ = VBG, we analyze each column of the addition, considering carryovers
between digits.
1. **Units Place**: ...
2. **Tens Place**: ...
3. **Hundreds Place**: ...
From the hundreds place equation, 2Q + c2 = V. Since V is a single digit, 2Q + c2 <= 9. This limits Q to 1-4.
Testing each possible value of Q:
- **Q=1**: V=2 or 3. Only V=2 yields a valid solution (Q=1, V=2, B=3, G=4).
- **Q=2**: V=4 or 5. Only V=4 yields a valid solution (Q=2, V=4, B=6, G=8).
- **Q=3**: No valid solutions due to contradictions in carryovers.
- **Q=4**: V=8 or 9. Only V=9 yields a valid solution (Q=4, V=9, B=3, G=7).
Each solution satisfies the equation and uniqueness constraints.
</tot>
<Answer>
{"Q": 1, "V": 2, "B": 3, "G": 4}, {"Q": 2, "V": 4, "B": 6, "G": 8}, {"Q": 4, "V": 9, "B": 3, "G": 7}
</Answer>
```

Figure 5: Illustration of ToT reasoning pattern with solving an Alphametic puzzle as an example.

