# OpenReview forum: "ToTRL: Unlock LLM Tree-of-Thoughts Reasoning Potential through Puzzles Solving"
_ICLR.cc/2026/Conference — ICLR 2026 Conference Withdrawn Submission_

### Official Review · Reviewer_6YF1 · 2025-10-28

**Soundness:** 2
**Presentation:** 3
**Contribution:** 3
**Rating:** 4
**Confidence:** 3

**Summary:**

The paper proposes Tree-of-Thoughts Reinforcement Learning (ToTRL), an on-policy RL framework designed to move an LLM from linear chain-of-thought (CoT) reasoning to tree-of-thoughts (ToT) reasoning. The method comprises three main components:

1. Policy optimization: The policy is trained using a clipped ratio objective, optionally augmented with a KL divergence term to a reference policy.

2. Two-stage training: Stage 1 (“no-thinking mode”) uses a special prompt template to suppress the model’s usual CoT trace and induce explicit ToT steps within markup tags. Stage 2 (“thinking mode”) trains the model in standard generation mode to internalize the learned ToT behaviors for inference.

3. Task-driven learning: The policy is trained via puzzle games that benefit from branching search, including 6×6 Sudoku and alphametic puzzles.

The resulting ToTQwen3‑8B model is evaluated on both in-distribution puzzles and several out-of-distribution logic tasks, using accuracy as the primary metric. Results show consistent improvements over several ~7–9B baselines.

**Strengths:**

S1: The paper addresses the well-known inefficiency and verbosity of long CoT reasoning by enabling branching exploration with a global perspective, aligning with prior work in ToT and graph-based reasoning.

S2: The use of a rule-based validator combined with an exact-match reward is straightforward to reproduce for puzzle tasks and eliminates dependence on human-labeled rationales, reflecting trends in O1/R1-style RL frameworks.

S3: Empirical results show that ToTQwen3‑8B achieves higher accuracy with fewer thinking tokens than the Qwen3‑8B baseline across multiple tasks, offering a notable practical advantage in efficiency and computational cost.

**Weaknesses:**

W1: Equation (1) resembles a PPO-style clipped objective with an optional KL term to a reference model. Calling it REINFORCE may obscure the actual optimization method used.

W2: The exact set-equality reward (Eq. 5) is brittle; success may hinge on precise formatting or extraction of answers, which could inflate performance or reduce reproducibility.

W3: Comparisons omit relevant search-based alternatives, including: (i) self-consistency over CoT traces, (ii) explicit ToT BFS/MCTS as in the original ToT paper, (iii) RAP (planning with MCTS), and (iv) TS-LLM (AlphaZero-style value-guided search).

W4: It is unclear whether all baselines were evaluated with identical token budgets, early-stop rules, and “thinking mode” support. Baselines lacking special thinking channels may be disadvantaged, making cross-model comparisons potentially unfair.

W5: The training tasks are restricted to puzzles with exact-rule validators. Real-world reasoning tasks (coding, planning, open-ended writing, tool use) often require partial credit, multi-step execution, debugging, environment interaction, or human judgment. Training only on puzzles may not stress the full range of reasoning, error exploration, and backtracking required in practical scenarios.

**Questions:**

Q1: Is Eq. (1) implemented as PPO with ε‑clipping? If so, why call it REINFORCE?

Q2: What is the rollout size n per prompt, sampling temperature/top‑p, and maximum tokens per “thinking” segment? How is the stop instruction inserted when the thinking budget is reached, and how is a partial tree summarized before answering?

Q3 Did you try partial‑credit rewards (e.g., counting correct solutions, Sudoku constraint satisfaction), or step‑level validators?

Q4: Did every baseline receive the same token budget and stop rule? Several baselines lack special “thinking modes”, how did you ensure fairness?

Q5: How does the method perform on open-ended reasoning tasks as described in W5, beyond puzzles?

---

> ### Author Response · Authors · 2025-11-21
>
> We thank the **Reviewer 6YF1** for their careful reading, positive feedback on the **efficiency and global perspective** of our method, and insightful questions that help us clarify the technical details and scope of our work.
>
> We address the weaknesses and questions below.
>
> ___
> > Weaknesses 1: Equation (1) resembles a PPO-style clipped objective with an optional KL term to a reference model. Calling it REINFORCE may obscure the actual optimization method used.
> > Question 1: Is Eq.(1) implemented as PPO with $\epsilon$-clipping? If so, why call it REINFORCE?
>
> Thanks for the sharp observation and for highlighting this discrepancy.
>
> You are entirely correct: the objective function in **Equation (1)** is a PPO-style clipped objective. We apologize for this confusion.
> Our use of the term "REINFORCE" inaccurately conflates the overall optimization method with a specific implementation detail, the advantage estimation technique shown in **Equation (2)**, which is commonly associated with REINFORCE.
>
> In **Section 2.1**, we correct this terminology and state clearly that we implement a **PPO** algorithm.
>
> ___
> > Weaknesses 2: The exact set-equality reward (Eq. 5) is brittle; success may hinge on precise formatting or extraction of answers, which could inflate performance or reduce reproducibility.
> > Question 3: Did you try partial-credit rewards (e.g., counting correct solutions, Sudoku constraint satisfaction), or step-level validators?
>
> We explore partial-credit rewards, and the experimental results (updated in **Appendix C** and **Table 10**) clarify our choice of the full-credit approach:
>
> * As the table below shows, using partial rewards causes the model to **converge prematurely** on strategies that output only a subset of correct answers. This ultimately **collapses the ToT thinking process** and yields lower overall scores compared to the full-credit approach.
> * Although step-level validators offer a denser reward signal, designing an **accurate and reliable intermediate reward** for each step in a branched, abstract reasoning trajectory is highly challenging without human labeling.
>
> |Rewards|5x5 Crossword|9x9 Sudoku|KK Puzzle|Poker 24|Make 24 Puzzle|
> |-|-|-|-|-|-|
> |partial-credit|0.200|0.040|0.836|0.238|0.315|
> |full-credit|**0.508**|**0.260**|**0.986**|**0.603**|**0.661**|
>
> ___
> > Weaknesses 3: Comparisons omit relevant search-based alternatives, including: (i) self-consistency over CoT traces, (ii) explicit ToT BFS/MCTS as in the original ToT paper, (iii) RAP (planning with MCTS), and (iv) TS-LLM (AlphaZero-style value-guided search).
>
> We agree that comparison to **explicitly guided search algorithms** is valuable.
>
> Our core contribution is to **activate a ToT exploration capability** within the LLM itself, as demonstrated by the prompt in Figure 1. Our method does not explicitly guide the model to use a specific search algorithm (e.g., BFS or MCTS); instead, it trains a policy that *learns* to explore.
>
> Explicitly guided search algorithms typically require **customized frameworks** for different tasks. Therefore, we **compare ToTRL with ToT and TS-LLM** on the Poker 24 Game. We use the Qwen3-8B-think for the ToT and TS-LLM.
>
> ||CoT|ToT|TS-LLM|ToTRL|
> |-|-|-|-|-|
> |Poker 24 Game|0.468|0.631|0.582|0.603|
>
> The results show that our ToTRL achieves **performance similar to** these explicitly guided tree-search methods, but with **greater practicality** because we require **only a single invocation** and **do not need a task-specific framework**. This demonstrates the strong utility of internalizing the ToT thinking process directly into the LLM.
>
> We **add these experimental results to Appendix E**.

---

> ### Author Response · Authors · 2025-11-21
>
> ___
> > Weaknesses 4: It is unclear whether all baselines were evaluated with identical token budgets, early-stop rules, and “thinking mode” support. Baselines lacking special thinking channels may be disadvantaged, making cross-model comparisons potentially unfair.
> > Question 4: Did every baseline receive the same token budget and stop rule? Several baselines lack special "thinking modes", how did you ensure fairness?
>
> * **Consistent Budget:** As detailed in **Section 3.1**, we set an identical maximum reply token length for every model and every task. This length is determined to allow for a **complete, non-repetitive reasoning trace** and **final answer**.
> * **Fair Comparison:** Baselines without an explicit "thinking mode" are permitted to use the **entire allocated token budget** for their reasoning and final answer. Our method, ToTQwen3-8B, also adheres to this maximum constraint. The resulting efficiency gain, where the trained policy *learns* to utilize the budget more effectively by exploring branches and converging to a solution **sooner**, is the key result of our work. This efficiency is achieved *under* the same maximum budget constraint, ensuring a fair comparison.
>
> ___
> > Weaknesses 5: The training tasks are restricted to puzzles with exact-rule validators. Real-world reasoning tasks (coding, planning, open-ended writing, tool use) often require partial credit, multi-step execution, debugging, environment interaction, or human judgment. Training only on puzzles may not stress the full range of reasoning, error exploration, and backtracking required in practical scenarios.
> > Question 5: How does the method perform on open-ended reasoning tasks as described in W5, beyond puzzles?
>
> We follow a common paradigm in which complex reasoning skills are first *activated* and trained on clear, rule-based tasks before being generalized to diverse scenarios:
>
> 1.  **Activation:** Our core contribution is to **activate the ToT exploration mechanism** using a suitable set of puzzle tasks with exact validators.
> 2.  **Generalization:** We focus our future work squarely on generalizing this learned exploration policy using RL to other practical, open-ended domains like coding and agentic planning.
>
> Current observations clarify the current scope:
>
> * **Programming/Coding:** We have not yet observed a statistically significant performance *loss* or *gain* in these tasks.
> * **Open-Ended Agentic Tasks:** We cannot make strong generalization claims in this paper for highly open-ended scenarios beyond the demonstrated tasks.
>
> We **provide our discussion in Appendix E** to more clearly present this viewpoint and emphasize that generalization to open-ended scenarios remains the most important direction for future research.
>
> ___
> > Question 2: What is the rollout size n per prompt, sampling temperature/top-p, and maximum tokens per "thinking" segment? How is the stop instruction inserted when the thinking budget is reached, and how is a partial tree summarized before answering?
>
> - As detailed in **Section 3.1**, we employed a batch size of 9, a rollout size of 16, and a maximum sequence length of 16384 tokens.
>
> - Following Qwen3, when the thinking output reaches this predefined limit, we **manually halt** the thinking process and insert a standardized stop-thinking instruction: `"Considering the limited time by the user, I have to give the solution based on the thinking directly now.\n</think>.\n\n"`. The model then generates a final response based on its accumulated reasoning up to that point.
> We provide a detailed explanation in the **Appendix B.3**.
>
> Moreover, we provide **demonstrative examples** of partial tree summarized before answering in **Appendix D** and **Figure 4 and 5** of the revised manuscript.
>
> [1] Yang, An, et al. "Qwen3 Technical Report." arXiv preprint arXiv:2505.09388, 2025.

---

> > ### Comment · Reviewer_6YF1 · 2025-11-24
> >
> > Thanks for the authors’ response. I appreciate the effort to address my questions. However, my main concern still lies in the nature of the evaluated puzzle tasks. While these tasks are suitable for demonstration, they are not common or representative of real-world scenarios, especially for LLMs to address. I encourage the authors to evaluate more realistic, real-world problems to better demonstrate the practical usefulness of the proposed approach.
> >
> > I will keep my current score.

---

> > > ### Author Response · Authors · 2025-11-25
> > >
> > > We sincerely thank the reviewer for highlighting the importance of evaluating our approach on more realistic, real-world scenarios.
> > >
> > > To address this, we evaluate **ToTQwen3-8B** on several realistic benchmarks, comparing it against the **Qwen3-8B (Thinking)** baseline. The results are presented below:
> > >
> > > | Benchmark | **BFCL v3 (Live)** | **LiveCodeBench v5** | **Arena Hard** | **Creative Writing v3** |
> > > |-|-|-|-|-|
> > > | Qwen3-8B (Thinking) | 0.767 | **0.556** | 0.832 | 0.727 |
> > > | ToTQwen3-8B (Ours) | **0.783** | 0.554 | **0.858** | **0.756** |
> > >
> > > As the results indicate, we observe modest performance improvements on agentic tool use (BFCL v3) and writing tasks (Arena Hard, Creative Writing), while no statistically significant difference has been observed so far on coding benchmarks (LiveCodeBench v5).
> > > These findings suggest that our approach holds preliminary yet promising potential for real-world applications.
> > > These initial findings motivate our future work, where we aim to focus on generalizing this learned exploration policy to other practical, complex domains such as coding and agentic planning utilizing RL.
> > >
> > > We have included these additional evaluations and a more detailed discussion of this viewpoint in **Appendix E**.

---

### Official Review · Reviewer_dHrF · 2025-10-31

**Soundness:** 2
**Presentation:** 2
**Contribution:** 2
**Rating:** 4
**Confidence:** 3

**Summary:**

The article introduce tree-of-thoughts RL (ToTRL) framework to guide LLMs to develop parallel ToT capabilities beyond sequential CoT. After ToTRL training process, the LLMs can solve puzzle games better, including in-domain and out-of-domain ones.

**Strengths:**

Introducing parallel thinking patterns into reasoning LLMs sounds a reasonable effort.

The ToTQwen3-8B model shows significant performance gains on a variety of logic puzzles.

**Weaknesses:**

Since the authors are still leveraging the CoT prompt for mathematical problems, it is unclear to me why it improves OOD mathematical tasks. Can you provide analysis as to why it also helps mathematical tasks?

I am particularly curious why ToTQwen3-8B can “explore the solution space more effectively and efficiently” as the authors mentioned, given that ToT is often very costly. Some experiment setting details of Figure 3 in section 3.5 are unclear. How do you set the budgets as exactly (2^c) k tokens? Do you set a budget for each method and truncate the thinking length?

**Questions:**

“Initially, as illustrated in Figure 1, the LLM undergoes training to perform ToT reasoning in a non-thinking mode. The non-reasoning mode is achieved by introducing blanks between reasoning tags, which compels the model to suspend its conventional reasoning processes.” This is not explained clearly, even after referring to Figure 1.

What does the separation line in Table 3 mean? Are there fundamental differences between the above 2 models and the middle 3 models?

The authors mention “Collectively, these efforts demonstrate the significant potential of internalizing ToT capabilities within the LLM itself, moving towards more autonomous reasoning.” I find the paper *Autonomous Tree-search Ability of Large Language Models* proposed the notion of autonomous ToT reasoning ability two years ago, and I believe there were relevant efforts in the literature. The authors can also consider including a discussion of the literature in this direction.

---

> ### Author Response · Authors · 2025-11-21
>
> We thank **Reviewer dHrF** for the thorough review and constructive feedback. We are encouraged by the reviewer's positive assessment of our work's effort in introducing parallel thinking patterns and the significant performance gains of ToTQwen3-8B.
>
> We address each concern and question below.
>
> ___
> >Weaknesses 1: Since the authors are still leveraging the CoT prompt for mathematical problems, it is unclear to me why it improves OOD mathematical tasks. Can you provide analysis as to why it also helps mathematical tasks?
>
> The ToT training process encourages the model to generate more diverse, parallel, and structured intermediate reasoning steps. Each parallel branch is thinking with the CoT format. This ultimately improves the quality of the sequential reasoning steps generated within the CoT prompt, which is crucial for tackling complex OOD problems.
> We provide a deeper discussion in **Section 3.4** of the revised manuscript.
>
>
> ___
> > Weaknesses 2: I am particularly curious why ToTQwen3-8B can "explore the solution space more effectively and efficiently" as the authors mentioned, given that ToT is often very costly. Some experiment setting details of Figure 3 in section 3.5 are unclear. How do you set the budgets as exactly (2^c) k tokens? Do you set a budget for each method and truncate the thinking length?
>
> While ToT is inherently slower than a fast CoT attempt, it achieves superior efficiency compared to alternative "slow thinking" methods (e.g., self-correction or extensive trial-and-error in unguided CoT). By leveraging a systematic ToT process, our model significantly reduces the **redundancy and wasted computation** associated with flawed or meaningless reasoning attempts. This leads to a more effective exploration of the solution space and, therefore, greater efficiency (less token consumption for successful problem-solving).
>
> We apologize for the initial lack of clarity. The budget definition is adopted from Qwen3[1] for fair comparison and refers to a **maximum token length threshold** for the model's thinking output. When the thinking output reaches this predefined limit, we **manually halt** the thinking process and insert a standardized stop-thinking instruction: `"Considering the limited time by the user, I have to give the solution based on the thinking directly now.\n</think>.\n\n"`. The model then generates a final response based on its accumulated reasoning up to that point.
> This ensures that all methods compared have an equivalent, constrained computational budget.
> We provide a detailed explanation in the **Appendix B.3**.
>
> [1] Yang, An, et al. "Qwen3 Technical Report." arXiv preprint arXiv:2505.09388, 2025.
>
> ___
> >Question 1: "Initially, as illustrated in Figure 1, the LLM undergoes training to perform ToT reasoning in a non-thinking mode. The non-reasoning mode is achieved by introducing blanks between reasoning tags, which compels the model to suspend its conventional reasoning processes." This is not explained clearly, even after referring to Figure 1.
>
> The "non-thinking mode" refers to the process shown in **stage 1** (bottom-left) of Figure 1. In this stage, the model is trained to generate the reasoning tags (`<think>` and `</think>`) with **no content** in between, resulting in an *empty* thinking block. This empty block compels the model to output the final answer *directly* without performing an overt internal reasoning search, thus contrasting with the detailed reasoning shown in the *thinking mode* (Stage 2).
>
> ___
> > Question 2: What does the separation line in Table 3 mean? Are there fundamental differences between the above 2 models and the middle 3 models?
>
> The separation line in **Tables 1, 2, and 3** is used to categorically distinguish the models based on their **training methodology**:
>
> * **Models above the separation line:** These models are primarily trained using SFT, typically distilling reasoning data from a stronger teacher model.
> * **Models below the separation line (the middle models):** These models have been further enhanced or trained using RL.

---

> ### Author Response · Authors · 2025-11-21
>
> ___
> >Question 3: The authors mention "Collectively, these efforts demonstrate the significant potential of internalizing ToT capabilities within the LLM itself, moving towards more autonomous reasoning." I find the paper *Autonomous Tree-search Ability of Large Language Models* proposed the notion of autonomous ToT reasoning ability two years ago, and I believe there were relevant efforts in the literature. The authors can also consider including a discussion of the literature in this direction.
>
> We appreciate the reviewer for suggesting this important citation and direction. We have updated the **Appendix A.1** section of the revised manuscript to include a discussion of the seminal work, *Autonomous Tree-search Ability of Large Language Models*, and other relevant efforts in the direction of internalized, autonomous deliberative reasoning, thereby better situating our contribution within the broader field.

---

### Official Review · Reviewer_NCNz · 2025-10-31

**Soundness:** 2
**Presentation:** 3
**Contribution:** 2
**Rating:** 2
**Confidence:** 4

**Summary:**

This paper proposed to apply RL to Tree-of-thoughts (ToT) with two-stage training, named ToTRL. This approach aims to narrow the gap between linear COT and parallel TOT generations, hence at the first stage the reasoning trace from the original model is turned off, then at the second stage both COT and TOT thoughts are turned on with RL for training. The paper tunes the model on two puzzle tasks for adaptation to the improved TOT reasoning patterns.

**Strengths:**

The paper tackles an important problem of how to design an effective training procedure of improving parallel thinking techniques like TOT.

**Weaknesses:**

- The motivation to adapt CoT to ToT reasoning is not well justified. It remains unclear in what sense is the linear COT unsuitable under the TOT setting, and whether the gain from 2-stage training is just due to extended-training.
- It doesn't seem to be convincing that by applying RL on only two puzzle tasks, the model performance can be improved over a wide range of reasoning tasks. The claim of the title that training on puzzle tasks can unlock the potential of ToT is very broad and needs deeper justification.
- The experimental result analysis did not reveal whether ToTRL truly improved tree search quality (diversity and depth).

**Questions:**

1. In the 2nd stage of ToTRL ("thinking mode"):
- How do thoughts between <think> and </think> differ from <tot> and </tot>? It seems what <tot> captures is just a summary of <think>, rather than novel thoughts that could further improve tree search quality.
- If thoughts between <think> and </think> come from the 1st stage ("no-thinking mode"), then since the base model did not go through ToT training yet, the thought quality is expected to be bad? This is exactly the problem the paper wants to address, not sure how effective it is to use these thoughts directly for 2nd stage training.

2. For fair comparisoin, it seems the baseline models in experiments should also adopt ToT style reasoning, rather than using on their native reasoning paths.

3. Was the improvement from ToTRL just an artifact of prolonged training, or it truly improved the quality of search trees? There isn't analysis on the difference between tree quality before and after ToTRL.

---

> ### Author Response · Authors · 2025-11-21
>
> We thank **Reviewer NCNz** for their time and comments. We are encouraged that the reviewer recognizes the importance of designing an effective training procedure for improving parallel thinking techniques.
>
> We address the weaknesses and questions below.
>
> ___
> >Weaknesses 1: The motivation to adapt CoT to ToT reasoning is not well justified. It remains unclear in what sense is the linear COT unsuitable under the TOT setting, and whether the gain from 2-stage training is just due to extended-training.
>
> **Linear CoT** is fundamentally unsuitable for exploring **multiple, branching intermediate steps**, which is a core requirement for complex problems like puzzles. A successful solution often necessitates exploring different **parallel paths**. **ToT**, by design, supports this parallel, tree-structured search, making it the superior reasoning paradigm for these tasks.
>
> The gain is not merely due to extended training. We conduct an additional ablation experiment comparing CoT and ToT under the **same total number of training steps**. The results below clearly show that the **ToT thinking paradigm provides a significant performance improvement** over the linear CoT paradigm. We add this experimental table to the **Appendix C** and **Table 11** for clarity.
>
> |Training Paradigm|5x5 Crossword|9x9 Sudoku|KK Puzzle|Poker 24|Make 24 Puzzle|
> |-|-|-|-|-|-|
> |CoT|0.404|0.180|0.957|0.526|0.628|
> |ToT|**0.508**|**0.260**|**0.986**|**0.603**|**0.661**|
>
> ___
> >Question 1: In the 2nd stage of ToTRL ("thinking mode"):
> - How do thoughts between [object Object] and [object Object] differ from [object Object] and [object Object]? It seems what [object Object] captures is just a summary of [object Object], rather than novel thoughts that could further improve tree search quality.
> - If thoughts between [object Object] and [object Object] come from the 1st stage ("no-thinking mode"), then since the base model did not go through ToT training yet, the thought quality is expected to be bad? This is exactly the problem the paper wants to address, not sure how effective it is to use these thoughts directly for 2nd stage training.
>
> This part of the review comment is not clearly rendered, as the placeholder text `[object Object]` makes the specific components the reviewer is asking about indiscernible.
> We kindly request that the reviewer **carefully re-examine and clarify their question** regarding the different thought components in the second stage of ToTRL ("thinking mode"). Once the specific parts are clearly identified, we will provide a detailed explanation.
>
> ___
> > Weaknesses 2: It doesn't seem to be convincing that by applying RL on only two puzzle tasks, the model performance can be improved over a wide range of reasoning tasks. The claim of the title that training on puzzle tasks can unlock the potential of ToT is very broad and needs deeper justification.
>
> As illustrate in **Section 2.4**, we choose puzzle tasks to **activate the ToT capability** because their inherent structure naturally necessitates the model to explore a **tree-like branching structure** of intermediate states. This makes them an ideal, low-barrier environment to bootstrap and reinforce the **ToT search strategy** in an LLM.
>
> We specifically select these two puzzle tasks because their difficulty is appropriately tuned for RL training on current-scale LLMs. Excessive difficulty prevents RL convergence, while minimal difficulty yields negligible improvement.
>
> Similar to how long-CoT reasoning is often activated and generalized via focused RL training on **mathematical tasks**, our contribution is to demonstrate that an appropriate task set (puzzles) can effectively **activate the ToT thinking mode**.
>
> We revise the manuscript to **more precisely present these views** in **Appendix E**, focusing the claim on "**activating the ToT thinking mode**" rather than on broad generalization in the current scope. We believe that once activated, the ToT strategy can be generalized to other suitable common scenarios in future work.

---

> ### Author Response · Authors · 2025-11-21
>
> ___
> > Weaknesses 3: The experimental result analysis did not reveal whether ToTRL truly improved tree search quality (diversity and depth).
>
> > Question 3: Was the improvement from ToTRL just an artifact of prolonged training, or it truly improved the quality of search trees? There isn't analysis on the difference between tree quality before and after ToTRL.
>
> As addressed in Weakness 1, the ablation study under the **same total training steps** confirms that the **ToT thinking paradigm yields significantly greater improvements**, proving that our gain is not an artifact of prolonged training.
>
> Identifying a precise, universally agreed-upon **quantitative metric** for LLM-generated tree search quality (diversity and depth) remains an open challenge.
>
> Instead of a quantitative metric, we provide **case studies** to the **Appendix D** and **Figure 4 and 5**. This analysis compares the crucial reasoning paths (key intermediate states) generated by the CoT and ToT paradigms. The comparison demonstrates that the performance uplift is directly attributed to the model's ability to **construct a more coherent and effective ToT reasoning path**.
>
> ___
> > Question 2: For fair comparisoin, it seems the baseline models in experiments should also adopt ToT style reasoning, rather than using on their native reasoning paths.
>
> We agree and conduct the suggested experiments. The results are presented as follows and are also added to the **Appendix C** and **Table 9**.
>
> When we guide the **untrained** baseline models to use the ToT reasoning strategy via prompting, the performance is **generally worse or highly inconsistent** compared to their native CoT reasoning. This finding strongly validates the core motivation of our work: **untrained LLMs cannot effectively utilize the ToT strategy via simple prompting**. Specialized training like ToTRL is essential to unlock and maximize this potential.
>
> |Model|5x5 Crossword|9x9 Sudoku|KK Puzzle|Poker 24|Make 24 Puzzle|
> |-|-|-|-|-|-|
> |DeepSeek-R1-Distill-Qwen-7B|0.000|0.000|0.012|0.010|0.000|
> |Llama-3.1-Nemotron-Nano-8B|0.000|0.000|0.032|0.047|0.033|
> |GLM-4-Z1-9B-0414|0.060|0.000|0.710|0.356|0.417|
> |Phi-4 Reasoning|0.000|0.040|0.722|0.054|0.232|
> |Qwen3-8B (Thinking)|0.376|0.080|0.700|0.485|0.600|
> |ToTQwen3-8B (Ours)|**0.508**|**0.260**|**0.986**|**0.603**|**0.661**|

---

### Official Review · Reviewer_R5RJ · 2025-10-31

**Soundness:** 3
**Presentation:** 4
**Contribution:** 4
**Rating:** 8
**Confidence:** 3

**Summary:**

This paper adds reinforcement learning to the tree-of-thoughts (which is a generalization of CoT) to support better LLM reasoning for solving games/puzzles/reasoning tasks. CoT is linear (ie sequential) and ToT can support branching exploration of multiple pathways. The authors' contribution is to add an on-policy RL on top of a rule-based reward system to help the LLM transition from sequential CoT to parallel, tree-structured reasoning.

**Strengths:**

- The paper is extremely well written and details are fleshed out to support reproducibility
- The experimental results show significant improvements based on a Qwen model that the authors have trained/fine-tuned.
- The authors also demonstrate their approach in a test-time-scaling experiment and show that the learned policy is good to explore the search space better.

**Weaknesses:**

- In the beginning of the paper, the authors mention that "Initially, the LLM is trained to perform ToT reasoning in a non-thinking mode, leveraging more moldable thinking patterns to activate ToT reasoning. Once the LLM has developed a degree of ToT reasoning ability in the non-reasoning mode, it undergoes
further training in the reasoning mode." This wasn't re-referred back later in the paper. Can you show/demonstrate examples of these patterns that activate ToT? Can you show ablation results showing the necessity of this initial reasoning in non-thinking mode? Is it because the CoTs are not "faithful"?

**Questions:**

- Please think of an additional experiment to address my question above.

---

> ### Author Response · Authors · 2025-11-21
>
> We thank the **Reviewer R5RJ** for recognizing the paper's **clarity, reproducibility, and the significant improvements** demonstrated by our approach, especially in the test-time scaling experiment.
>
> We address the weaknesses and questions below.
>
> ___
> >Weaknesses: In the beginning of the paper, the authors mention that "Initially, the LLM is trained to perform ToT reasoning in a non-thinking mode, leveraging more moldable thinking patterns to activate ToT reasoning. Once the LLM has developed a degree of ToT reasoning ability in the non-reasoning mode, it undergoes further training in the reasoning mode." This wasn't re-referred back later in the paper. Can you show/demonstrate examples of these patterns that activate ToT? Can you show ablation results showing the necessity of this initial reasoning in non-thinking mode? Is it because the CoTs are not "faithful"?
>
> We confirm that the necessity of this initial training stage (stage 1) is demonstrated by the **ablation study** presented in **Section 3.6** and **Table 6 (Ablation study on multi-stage ToTRL)**. The results clearly show that **excluding stage 1 training consistently leads to a performance drop** across all evaluated tasks. Stage 1 is crucial because it **instills the foundational ToT structure** before introducing the more complex **reasoning-mode** (stage 2) objectives.
>
> |Stage1|Stage2|5x5 Crossword|9x9 Sudoku|KK Puzzle|Poker 24|Make 24 Puzzle|
> |-|-|-|-|-|-|-|
> ||✔|0.470|0.160|0.957|0.485|0.532|
> |✔|✔|**0.508**|**0.260**|**0.986**|**0.603**|**0.661**|
>
> Moreover, we provide **demonstrative examples** of these foundational patterns, which activate the core ToT reasoning ability, in **Appendix D** and **Figure 4 and 5** of the revised manuscript.

---

### Note · Authors · 2025-12-13

I have read and agree with the venue's withdrawal policy on behalf of myself and my co-authors.